# Battle of the Large Language Models: Dolly vs LLaMA vs Vicuna vs Guanaco vs Bard vs ChatGPT - A Text-to-SQL Parsing Comparison

Shuo Sun[1], Yuchen Zhang[1,2], Jiahuan Yan[3], Yuze Gao[1], Donovan Ong[*,4], Bin Chen[1], Jian Su[1]

[1]Institute for Infocomm Research (I[2]R), A*STAR, Singapore
[2]CNRS@CREATE LTD, Singapore, [3]National University of Singapore
[4]Nanyang Technology University

[1]{Sun_Shuo,Zhang_Yuchen,Gao_Yuze,Donovan_Ong,bchen,sujian}@i2r.a-star.edu.sg,
[3]jiahuan_22@u.nus.edu, [4]s21006@e.ntu.edu.sg

## Abstract

The success of ChatGPT has ignited an AI race, with researchers striving to develop new large language models (LLMs) that can match or surpass the language understanding and generation abilities of commercial ones. In recent times, a number of models have emerged, claiming performance near that of GPT-3.5 or GPT-4 through various instruction-tuning methods. As practitioners of Text-to-SQL parsing, we are grateful for their valuable contributions to open-source research. However, it is important to approach these claims with a sense of scrutiny and ascertain the actual effectiveness of these models. Therefore, we pit six popular large language models against each other, systematically evaluating their Text-to-SQL parsing capability on nine benchmark datasets with five different prompting strategies, covering both zero-shot and few-shot scenarios. Regrettably, the open-sourced models fell significantly short of the performance achieved by closed-source models like GPT-3.5, highlighting the need for further work to bridge the performance gap between these models.

## 1 Introduction

Text-to-SQL parsing automatically converts user input questions into SQL statements, enabling the retrieval of relevant information from databases. By enabling users to express their objectives in natural language, Text-to-SQL systems could minimize the technical obstacles for non-expert users to interact with relational databases and enhance productivity.

The introduction of large pre-trained language models like BERT (Devlin et al., 2019) and T5 (Raffel et al., 2020) has further improved the performance of Text-to-SQL systems. Researchers have been leveraging the impressive understanding of these models to push the boundaries of Text-to-SQL capabilities.

Recently, breakthroughs in decoder-based large language models (Brown et al., 2020b; Touvron et al., 2023) have further revolutionized the field of NLP. A prominent trend is the pursuit of training increasingly larger language models, encompassing billions of parameters, and utilizing vast amounts of textual data. Subsequently, these models are fine-tuned using instruction-based techniques, enabling them to better adhere to human-generated text prompts.

Among the prominent applications of decoder LLMs is ChatGPT, which is built upon OpenAI's GPT-3.5 and GPT-4 models. ChatGPT has demonstrated exceptional capabilities in zero-shot and few-shot scenarios, as evidenced by various Text-to-SQL evaluation studies (Rajkumar et al., 2022; Liu et al., 2023). Regrettably, the success of Chat-GPT has ignited an AI race, leading industry research labs to cease public disclosure of their model parameters and training methodologies.

Therefore, researchers have been actively pursuing the development of new language models that can potentially rival the capabilities of Chat-GPT. These models include Dolly, which builds upon the Pythia models (Biderman et al., 2023), as well as Vicuna (Chiang et al., 2023) and Guanaco (Dettmers et al., 2023), based on the LLaMA models (Touvron et al., 2023). Some of them have garnered attention by claiming to achieve performance levels surpassing 90% of GPT-4 through fine-tuning techniques.

As practitioners of Text-to-SQL, we are grateful for the contributions made by these models. However, we remain uncertain about whether these open-source models truly deliver the level of quality they claim to achieve. To address this concern, this paper presents a comprehensive evaluation of

---

*Work done when Donovan was working at I[2]R

six language models: Dolly, LLaMA, Vicuna, Guanaco, Bard, and ChatGPT, directly comparing their performance on nine benchmark datasets, utilizing five distinct prompting strategies.

Our main findings are:

1. Open-source models demonstrate notably **inferior performance** compared to closed-source models across a majority of Text-to-SQL datasets.

2. While LLMs demonstrate proficiency in generating syntactically valid SQL statements, they often struggle to produce semantically accurate queries.

3. LLMs prove to be highly sensitive to the examples utilized for few-shot learning.

To facilitate further research in the field of Text-to-SQL parsing, we are making all raw and post-processed outputs from the large language models publicly at `https://github.com/ZhangYuchenYC/deepeval-Text2SQL`.

## 2 Experiment Setup

### 2.1 The Large Language Model Contestants

We have carefully selected contestants from five families of large language models to provide a comprehensive representation of the current landscape in the field:

**Dolly**[1] is a 12 billion parameter language model, claimed to be the first publicly available instruction-tuned LLM licensed for both academic and commercial applications. It is based on Pythia (Biderman et al., 2023) and has undergone fine-tuning using an instruction dataset created by Databricks employees. Additionally, we have experimented with smaller variants of Dolly, including the 3B and 7B versions.

**LLaMA** (Touvron et al., 2023) is a collection of large language models ranging from 7 billion to 65 billion parameters. These models have been trained exclusively on publicly available text corpora. Unlike the other models, LLaMA models are not instruction fine-tuned. To ensure the natural continuation of prompts, we append the keyword "SELECT" at the end of the prompts when interacting with LLaMA models.

**Vicuna** (Chiang et al., 2023) is a 13 billion parameter LLaMA model fine-tuned on user-shared

conversations collected from ShareGPT[2], which claims to achieve 90% ChatGPT quality based on an automated evaluation with GPT-4. We also experiment with the 7B version.

**Guanaco** (Dettmers et al., 2023) is a family of large language models claiming to achieve 99.3% of ChatGPT performance with only 24 hours of fine-tuning on one GPU. Similar to Vicuna, the Guanaco model was instruction-tuned on the LLaMA models. We conduct our evaluations on the 33B version.

**Bard**[3] is a conversational chatbot released by Google as an answer to OpenAI's chatGPT. It is initially powered by LaMDA (Thoppilan et al., 2022) and later transitioned to PaLM 2 (Anil et al., 2023). As the technical details of Bard are not publicly released, we evaluate its performance on Text-to-SQL datasets as a black box model. We denote the version powered by LaMDA as **Bard-L** and the version powered by PaLM 2 as **Bard-P2**.

**GPT-3.5** (Brown et al., 2020a) is OpenAI's most cost-effective large language model optimized for chat-based applications at the point of writing this paper. It has 175 billion parameters and powers the popular ChatGPT chatbot. We conduct all evaluations on the "GPT-3.5-turbo-0301" variant of GPT-3.5 through openAI's API.

### 2.2 Prompting Strategies

We explore five commonly used prompting strategies:

The **Informal Schema (IS)** strategy provides a description of tables and their associated columns in natural language. In this approach, the schema information is expressed in a less formal manner. In contrast, the **API Docs (AD)** strategy, as outlined in the evaluation conducted by Rajkumar et al. (2022), follows the default SQL translate prompt provided in OpenAI's documentation[4]. This prompt adheres to a slightly more formal definition of the database schema. The **Select 3** strategy includes three example rows for each table in the database. This additional information aims to provide concrete examples of the data contained within each table, supplementing the schema description. We also investigate the effectiveness of **1-Shot Learning (1SL)** and **5-Shot Learning (5SL)** strategies, where we provide one and five golden examples

---

[1]https://github.com/databrickslabs/dolly

[2]https://sharegpt.com/
[3]https://bard.google.com/
[4]https://platform.openai.com/examples/default-sql-translate

in the prompts, respectively. Examples of various prompting strategies can be found in the appendix.

## 2.3 Benchmark Datasets

| Dataset | Number of examples |
|---|---|
| Academic | 196 |
| ATIS | 347 |
| GeoQuery | 182 |
| Yelp | 128 |
| IMDB | 131 |
| Restaurants | 378 |
| Scholar | 315 |
| Advising | 1832 |
| Spider | 1034 |

Table 1: Number of examples used for evaluation of various Text-to-SQL datasets.

We evaluate the prompting strategies on nine Text-to-SQL datasets: **Academic** (Li and Jagadish, 2014), **ATIS** (Price, 1990; Dahl et al., 1994), **Geo-Query** (Zelle and Mooney, 1996), **Yelp** and **IMDB** (Yaghmazadeh et al., 2017), **Restaurants** (Tang and Mooney, 2000; Popescu et al., 2004), **Scholar** (Iyer et al., 2017), **Advising** (Finegan-Dollak et al., 2018) and **Spider** (Yu et al., 2018). Note that for the first eight datasets, we employ the standardized and improved versions released by Finegan-Dollak et al. (2018).

Following Zhong et al. (2020), we evaluate the performance of large language models on the test sets of the datasets if such sets were defined. In cases where test sets were not provided, we evaluated the models on the entire datasets. For the Spider dataset, we evaluate the models on the development set since the test set is not publicly available at the point of writing this paper. To maintain consistency with Zhong et al. (2020), we collectively refer to the first eight datasets as the *"classical datasets"*.

**Important details on data splits** Classical Text-to-SQL datasets such as GeoQuery, ATIS and Scholar often utilize a **question-based data split** strategy where matching (text, SQL) pairs are assigned to the same split. However, one potential issue with this split strategy is that the same SQL query could appear in both the training and testing set. This duplication of SQL queries across the training and testing sets can introduce a bias and potentially inflate the model's performance.

The model may inadvertently learn to memorize the specific SQL queries rather than understanding the underlying semantic parsing. Therefore, we employ the **query-based data splits** described in Finegan-Dollak et al. (2018) where SQL queries with similar structures are assigned to the same splits. As a result, some of our reported results are lower than those documented in previous work such as Rajkumar et al. (2022).

We also want to highlight that unlike Suhr et al. (2020); Lan et al. (2023) who use the filtered combinations of train and dev splits of GeoQuery, Scholar and Advising and the filtered dev split of ATIS as their evaluations sets, we adhere to the evaluation splits in Finegan-Dollak et al. (2018). The reason behind this decision was our intention to sample examples from the train sets for the purpose of conducting one-shot and five-shot prompting experiments.

## 2.4 Evaluation Metrics

The primary evaluation metric employed in this paper is the **execution accuracy (EX)**, which measures the percentage of generated SQL queries that precisely align with the outputs of the gold SQL queries. Additionally, for the Spider dataset, we also calculate the **test suite accuracy (TS)**, which serves as the official evaluation metric for this dataset. TS provides an upper-bound estimation of semantic accuracy by assessing the execution accuracy of predicted queries on a set of distilled randomly generated databases (Zhong et al., 2020).

Similar to Liu et al. (2023), we refrain from utilizing the exact match accuracy (Yu et al., 2018) metric as a SQL query can often be expressed in multiple equivalent ways to achieve the same objective. Consequently, exact match accuracy may inadvertently penalize large language models that generate SQL queries that differ in style from the gold data.

## 2.5 Evaluation Details

We utilize several models in our study, including three variants of Dolly (v2-3b, v2-7b, and v2-12b), two variants of Vicuna (7B and 13B), one variant of Guanaco (33B), and four variants of LLaMA (7B, 13B, 30B, and 65B). To ensure consistency, we aim to adhere closely to the default hyperparameters of each model. We set a top-p sampling rate of 0.92 and a temperature of 0.8, for Dolly, a temperature of 0.8 for Vicuna and Guanaco and top-p sampling rate of 0.95 and a temperature of 0.8 for

LLaMA. During the evaluation, we conduct our experiments on a server equipped with eight NVIDIA RTX A6000 GPUs. For Bard, we have developed a script that directly extracts evaluation outputs from its web user interface. For GPT3.5, we leverage the "gpt-3.5-turbo-0301" version through OpenAI's API and adhere to the default hyperparameters of a temperature of 1.0 and a top-p sampling rate of 1.0.

## 3 Evaluation Results

### 3.1 Spider Dataset

The accuracy of execution (EX) and the accuracy of the test suite (TS) on various combinations of prompting strategies and models are presented in Table 2. Our main findings are:

**Closed-source models exhibit superior performance compared to open-source models:** GPT-3.5 is the leading model, surpassing the second-place Bard model by 17.8% in execution accuracy (EX) and by 14.1% in test suite accuracy (TS). However, GPT-3.5 still falls behind state-of-the-art Text-to-SQL models like the one proposed by Li et al. (2023a) by a margin of at least 13% in absolute terms. Bard-P2 demonstrates some enhanced performance over Bard-L when employing the IS, S3, and 5SL prompting strategies, but suffers from significant performance degradations when using the AD and 1SL prompting strategies.

**Open-source models struggle with the Spider dataset:** Despite a positive correlation between the number of parameters and model performance, open-source models face challenges in achieving high accuracies on the Spider dataset. For example, although Vicuna 7B and 13B have demonstrated improvements over the raw pre-trained LLaMA 7B and 13B models, there still exists a significant gap in performance when compared to Bard and GPT-3.5. Furthermore, the Dolly models also demonstrate underperformance when compared to LLaMA's 13B version across different prompting strategies.

**The performance of LLMs are highly sensitive to the styles of prompts:** Our empirical findings confirm that there is no universal prompting strategy that works well across all models. While the IS prompting strategy proves effective for GPT-3.5, Bard, Vicuna, and Guanaco, it yields suboptimal accuracies for Dolly and LLaMA. Surprisingly, LLaMA achieves its optimal results when employing the S3 prompts, which in contrast, significantly

deteriorates the performance of GPT-3.5.

**Few-shot learning with random examples offers limited performance gains:** The majority of results obtained from 1SL and 5SL tend to underperform or, at best, achieve comparable results to other prompting strategies. However, there are a couple of exceptions to this trend. One exception is the Dolly model, which shows improved performance with the 1SL prompting strategy compared to other prompting strategies in the 12B variant. This result appears to be anomalous because similar gains in performance are not observed in other 1SL and 5SL results. The other exception is the LLaMA model, where the few-shot prompting strategies outperform some of the zero-shot strategies. For example, the 30B LLaMA model achieves 22.4% EX and 19.9% TS accuracy with only 5 given examples, which is close to the performance of the Guanaco model (24.4% EX and 19.0% TS).

### 3.2 Classical Datasets

Since there are no training sets for Academic, Restaurants, IMDB and Yelp, we sample examples for 1SL and 5SL from the evaluation sets of other classical datasets. We highlight some of our key findings based on the results in Table 3:

**LLMs show lacklustre performance on most of the classical datasets:** In particular, there is a noticeable disparity in the results obtained from the Academic and Restaurants datasets when compared to the baseline performance reported in previous research. The highest achieved accuracies on these datasets are merely 2.9% and 2.4% respectively, which is significantly lower than the baseline results of 34.0% and 45.2% observed in other studies utilizing traditional seq2seq models with LSTM or BERT (Devlin et al., 2019). Furthermore, even with instruction tuning, Vicuna, Guanaco and Dolly face considerable challenges on the classical datasets. They often yield nearly zero execution accuracies across various prompt strategies and datasets combinations.

**The effectiveness of few-shot learning varies across different models:** In contrast to the findings from the Spider dataset, we observe some performance improvements on LLaMA and GPT-3.5 with 1SL and 5SL. For example, the performance of GPT-3.5 on the GeoQuery dataset improves from 15.4% to 42.3% with 1SL, while the performance of LLaMA on the same dataset also significantly improves from 12.1% to 15.4% with 5SL. However,

| Models | #p | Prompting Strategies | | | | | | | | | |
| | | **IS** | | **AD** | | **S3** | | **1SL** | | **5SL** | |
| | | EX | TS | EX | TS | EX | TS | EX | TS | EX | TS |
| Dolly | 3B | 8.7 | 5.5 | 2.2 | 1.5 | 0.2 | 0.2 | 1.4 | 0.8 | 0.6 | 0.2 |
| | 7B | 9.2 | 6.5 | 0.5 | 0.3 | 0.6 | 0.4 | 1.7 | 1.3 | 0.4 | 0.3 |
| | 12B | 9.5 | 7.4 | 0.1 | 0.1 | 0.8 | 0.6 | **11.8** | **9.3** | 5.3 | 3.7 |
| LLaMA | 7B | 4.1 | 2.1 | 2.9 | 2.3 | 11.3 | 7.4 | 7.5 | 6.0 | 9.0 | 8.0 |
| | 13B | 8.7 | 4.8 | 6.1 | 4.4 | 16.2 | 12.8 | 13.5 | 11.4 | 14.4 | 13.2 |
| | 30B | 4.7 | 3.3 | 10.4 | 7.5 | 18.5 | 13.9 | 18.8 | 15.5 | 22.4 | 19.9 |
| | 65B | 12.2 | 9.1 | 14.0 | 10.5 | **29.5** | **23.9** | 23.7 | 19.2 | 23.8 | 20.1 |
| Vicuna | 7B | 25.8 | 19.6 | 17.6 | 13.8 | 18.2 | 13.8 | 16.7 | 13.2 | 5.0 | 3.7 |
| | 13B | **34.8** | **26.5** | 21.2 | 16.8 | 9.5 | 6.5 | 19.1 | 14.5 | 18.8 | 15.8 |
| Guanaco | 33B | **24.4** | **19.0** | 14.5 | 10.6 | 15.8 | 12.9 | 10.3 | 7.9 | 2.0 | 1.6 |
| Bard-L | UNK | 53.6 | 46.6 | 52.5 | 45.1 | 53.1 | 45.6 | 50.5 | 43.9 | 51.8 | 45.0 |
| Bard-P2 | | **60.2** | **52.3** | 48.7 | 41.8 | 54.6 | 46.1 | 47.8 | 41.4 | 53.6 | 46.9 |
| GPT-3.5 | 175B | **70.9** | 59.4 | 67.2 | 57.9 | 31.1 | 27.0 | 67.5 | 58.2 | 70.4 | **59.7** |

Table 2: Execution accuracy (EX) and test suite accuracy (TS) results for various large language models and prompting strategies (Informal Schema (IS), API Docs (AD), Select 3 (S3), 1-shot learning (1SL) and 5-shot learning (5SL)) on the Spider development set. The number of parameters (#p) for Bard is unknown (UNK). Highlighted in bold are the best results in each LLM family.

we do not see similar performance improvements with 1SL or 5SL for Dolly, Vicuna and Bard.

**Appending database example rows is ineffective:** Just like the outcomes observed with the spider dataset, the S3 prompting strategies yield subpar results when applied to the classical datasets across different models. Therefore, it is evident that the S3 prompting strategy may not be effective in the context of Text-to-SQL.

## 4 Discussions

### 4.1 Are LLMs generating valid SQLs?

One potential explanation for the underwhelming performance of large language models lies in their inability to grasp the intention behind prompts aimed at generating SQL statements. For instance, in response to the question "return me the homepage of PVLDB," Guanaco directly provided the answer "The website is https://www.vldb.org/pvldb." When faced with many S3 prompts, GPT-3.5 fails to generate valid responses. To assess the extent of such instances, we plotted the proportion of valid SQL statements generated using different prompt strategies for various large language models in Figure 1a and 1b.

For spider dataset, we discovered that many mod-els, excluding Dolly, consistently generate valid SQL responses over 90% of the time with IS, 1SL, and 5SL prompt strategies. Interestingly, LLaMA also demonstrates the ability to generate valid SQL statements, even though it was not specifically fine-tuned on instruction datasets. For the classical datasets, Bard-P2 and GPT-3.5 are still capable of generating valid SQLs in the 80-100% range. However, the open-source models such as Vicuna and Dolly encounter challenges in achieving a valid SQL percentage above 75%. What's particularly noteworthy is the divergent trends observed in LLaMA and Guanaco. LLaMA generates more valid SQLs through few-shot learning, whereas Guanaco's performance declines as the number of examples increases.

Furthermore, we noticed that the AD and S3 prompting strategies are generally suboptimal, as they lead to significant decreases in the number of valid SQL responses across all datasets for many large language models. GPT-3.5 is particularly susceptible to the S3 prompting strategy, resulting in a sharp decline in the percentage of valid SQLs generated in both the spider and classical datasets.

Lastly, it is crucial to emphasize that although these language models can produce valid SQL re-

| Model | #P | PS | Acad | ATIS | Adv | Geo | IMDB | Rest | Sch | Yelp | AVG |
|---|---|---|---|---|---|---|---|---|---|---|---|
| Dolly 2.0 | 12B | IS | 0.0 | 0.0 | 0.0 | **8.2** | **3.8** | 0.0 | 0.0 | **3.1** | **1.9** |
| | | AD | 0.0 | 0.0 | 0.0 | 0.0 | 0.0 | 0.0 | 0.0 | 0.0 | 0.0 |
| | | S3 | 0.0 | 0.0 | 0.0 | 0.0 | 1.5 | 0.0 | 0.0 | 0.8 | 0.3 |
| | | 1SL | **0.5** | 0.0 | 0.0 | 0.0 | 0.0 | 0.0 | 0.0 | 2.3 | 0.4 |
| | | 5SL | **0.5** | 0.0 | 0.0 | 0.0 | 0.8 | 0.0 | 0.0 | 0.8 | 0.3 |
| LLaMA | 30B | IS | 0.0 | 0.3 | 0.0 | 0.5 | 0.0 | 0.0 | 0.0 | 0.0 | 0.1 |
| | | AD | 0.0 | 0.0 | 0.0 | 12.1 | **3.1** | 0.0 | 0.3 | 1.6 | 2.1 |
| | | S3 | **1.0** | 0.3 | 0.0 | 2.7 | 0.0 | 0.0 | 0.3 | 1.6 | 0.7 |
| | | 1SL | 0.0 | 0.0 | 0.0 | 10.4 | 0.8 | 0.0 | 0.3 | 0.0 | 1.4 |
| | | 5SL | 0.5 | **0.6** | **0.1** | **15.4** | 0.0 | 0.0 | **2.5** | 1.6 | **2.6** |
| Vicuna | 13B | IS | **2.0** | 0.0 | 0.0 | 0.5 | **6.1** | 0.0 | 0.3 | 2.3 | 1.4 |
| | | AD | 0.0 | 0.0 | 0.0 | 1.6 | 4.6 | 0.0 | 0.3 | 1.6 | 1.0 |
| | | S3 | 1.0 | 0.0 | 0.0 | **4.9** | 3.1 | 0.0 | **1.0** | 2.3 | **1.5** |
| | | 1SL | 0.0 | **0.3** | **0.1** | 4.9 | 0.0 | 0.0 | 0.0 | 0.8 | 0.8 |
| | | 5SL | 0.0 | **0.3** | **0.1** | 2.7 | 0.0 | 0.0 | 0.3 | 0.8 | 0.5 |
| Guanaco | 33B | IS | 0.5 | **0.9** | 0.0 | 1.1 | **1.5** | 0.0 | 0.0 | **2.3** | 0.8 |
| | | AD | **2.0** | 0.6 | 0.0 | **2.7** | 0.8 | 0.0 | 0.0 | 0.8 | **0.9** |
| | | S3 | 0.0 | 0.0 | 0.0 | 0.0 | 0.0 | 0.0 | 0.0 | 0.0 | 0.0 |
| | | 1SL | 0.0 | 0.0 | 0.0 | 0.5 | 0.0 | 0.0 | **0.3** | 0.0 | 0.1 |
| | | 5SL | 0.0 | 0.0 | 0.0 | 0.5 | 0.0 | 0.0 | 0.0 | 0.0 | 0.1 |
| Bard-P2 | UNK | IS | **6.6** | 0.0 | 0.0 | 8.2 | **17.6** | 0.0 | 0.3 | 3.9 | **4.6** |
| | | AD | 5.6 | 0.3 | 0.0 | 7.7 | 11.5 | 0.0 | 0.3 | 3.1 | 3.6 |
| | | S3 | 3.1 | **1.2** | 0.0 | 4.4 | 7.6 | 0.0 | 0.3 | **6.2** | 2.9 |
| | | 1SL | 3.1 | 0.9 | 0.0 | **13.2** | 12.2 | 0.0 | **1.0** | 3.9 | 4.3 |
| | | 5SL | 1.0 | 0.9 | 0.0 | 3.8 | 10.7 | 0.0 | 0.0 | 2.3 | 2.3 |
| GPT-3.5 | 175B | IS | **13.8** | 1.4 | 0.0 | 15.4 | 17.6 | **2.4** | 2.2 | 4.7 | 7.2 |
| | | AD | 10.7 | 0.9 | 0.0 | 11.5 | 16.0 | 0.8 | 1.0 | 2.3 | 5.4 |
| | | S3 | 1.5 | 0.0 | 0.0 | 3.3 | 5.3 | 0.0 | 0.0 | 1.6 | 1.5 |
| | | 1SL | 11.2 | 2.6 | **0.3** | **42.3** | **18.3** | 2.4 | **6.3** | 6.2 | **11.2** |
| | | 5SL | 6.1 | **2.9** | **0.3** | 39.6 | 16.8 | 1.6 | 3.5 | **10.2** | 10.1 |
| text2sql-data | | - | **75.0** | **34.0** | **8.0** | **49.0** | 24.0 | 33.0 | 6.0 | 32.0 | **32.6** |
| XSP | | - | 12.1 | - | - | - | 33.3 | **45.2** | - | **49.2** | - |
| Unite | | - | - | - | - | - | **41.1** | - | - | - | - |

Table 3: EX and TS results for various LLMs and prompting strategies on classical datasets: **Acad**emic, **ATIS**, **Adv**ising, **Geo**Query, **IMDB**, **Rest**aurant, **Sch**olar and **Yelp**. We also include baseline results in Finegan-Dollak et al. (2018) (text2sql-data), XSP Suhr et al. (2020) (XSP) and Lan et al. (2023) (Unite) if possible. Highlighted in bold are the best results in each LLM family.

sponses, these SQLs are often semantically inaccurate and fail to adequately address the input text questions. As a consequence, the execution accuracies across the majority of datasets are notably low.

## 4.2 How does sample selection affect the performance of 1SL and 5SL?

Based on the results presented in Table 2 and Table 3, it becomes evident that the inclusion of random examples from the training set in prompts does not significantly enhance the performance of different models. The only exceptions are LLaMA and GPT-3.5, which demonstrate noticeable improvements

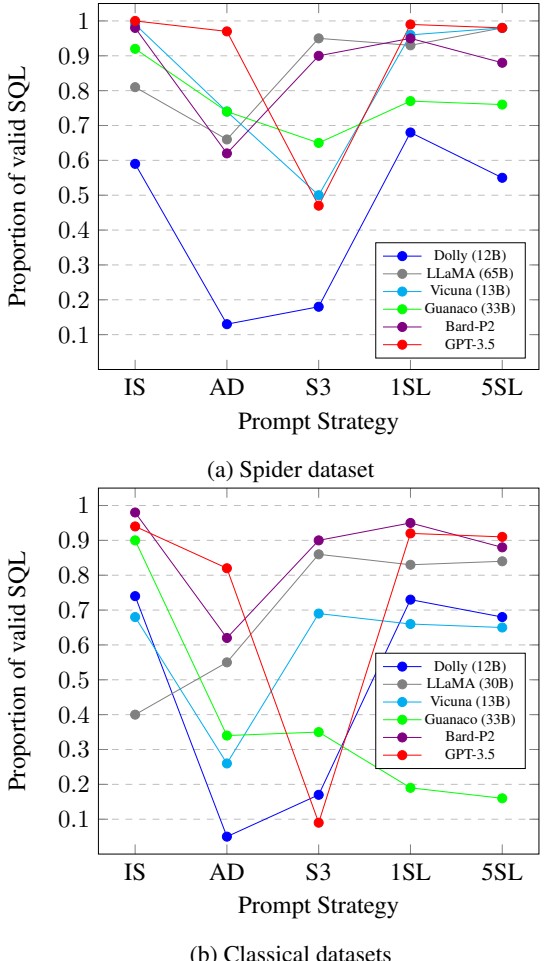

(a) Spider dataset

(b) Classical datasets

Figure 1: The proportion of valid SQL generated for different prompting strategies across multiple models. Figure (a) presents the results on Spider, while Figure (b) presents the average results based on the classical datasets.

across most classical datasets when utilizing 1SL and 5SL prompting strategies. The improvements in LLaMA's performance with 1SL or 5SL prompting strategies can be partially attributed to the fact that exposing LLaMA to more examples substantially enhances its ability to generate valid SQL, as depicted in Figure 1b.

**LLMs adapt to the canonicalized SQL style:** Another noteworthy observation is that when large language models are presented with examples from classical datasets, they start to generate SQL in a style similar to the canonicalized format described in Finegan-Dollak et al. (2018), as shown in Figure 2, where tables alias follow a standardized convention of <TABLE_NAME>alias<N>.

**Sensitivity of LLMs to style change:** To assess the extent to which Language Models (LLMs) follow the canonicalized SQL style while generating

```
SELECT PUBLICATIONalias0.ABSTRACT FROM PUBLICATION
AS PUBLICATIONalias0 WHERE PUBLICATIONalias0.TITLE
= "Making database systems usable";
```

Figure 2: Example of a canonicalized SQL statement.

SQLs with 1SL and 5SL, we examine the proportion of generated SQL statements that contain the term "alias" in Table 4. Our findings reveal that the change in generated SQL style is only evident when employing the 1SL and 5SL prompting strategies. Notably, LLaMA stands out among all the models, as it consistently appends the term "alias" to over 86% of the generated SQL statements. Interestingly, Bard is less sensitive to the canonicalized SQL style, with a style change observed in only 16.0% of all generated SQLs. On the other hand, GPT-3.5 demonstrates higher sensitivity, with more than 50% of the generated SQLs being affected. Based on this observation, we hypothesize that this disparity in sensitivity could be a contributing factor to the greater success of the 1SL and 5SL prompting strategies employed by LLaMA and GPT-3.5.

| Model | AD | IS | S3 | 1SL | 5SL |
|---|---|---|---|---|---|
| Dolly (12B) | 0.1 | 0.0 | 0.0 | 37.8 | 38.0 |
| LLaMA (30B) | 0.0 | 0.1 | 0.1 | 86.1 | 92.0 |
| Vicuna (13B) | 0.0 | 0.1 | 0.1 | 37.8 | 38.5 |
| Guanaco (33B) | 0.0 | 0.1 | 0.2 | 1.8 | 1.8 |
| Bard-P2 | 0.0 | 0.0 | 0.0 | 16.0 | 28.3 |
| GPT-3.5 | 0.0 | 0.0 | 0.0 | 51.6 | 58.3 |

Table 4: Percentage of generated SQLs containing the word "alias" for classical datasets.

**Impact of sampling from different sources on Performance**

We conclude this section by providing a brief discussion on experiments involving the sampling of examples from sources other than the training sets. Table 5 presents the 1SL and 5SL results obtained when samples are taken from two different sources: 1) the Spider train set, and 2) the evaluation sets. In the second case, we take precautions to avoid any potential answer leakage, by filtering out all examples that have the same SQL answer as the question of interest. We find that using examples from the Spider dataset not only fails to yield any benefits but also leads to a decline in the perfor-

mance of the models, performing worse than the zero-shot methods. On the other hand, when we include examples from the evaluation sets, we observe improvements in the evaluation results. Upon closer examination of the prompts, we discovered instances where the few-shot examples were syntactically similar to the expected SQL responses, differing primarily in terms of tables, columns, and values. This finding highlights the sensitivity of LLMs to the examples provided in the prompt. We hypothesize that LLMs may generate more accurate SQL statements if we feed them with examples that are syntactically close to the expected SQL response.

| Model | Train | Spider | Eval |
|---|---|---|---|
| Dolly (12B) | **1.5/0.5** | 0.4/0.4 | 0.8/0.3 |
| LLaMA (30B) | 1.4/2.6 | 1.0/1.3 | **8.2/17.8** |
| Vicuna (13B) | 0.8/0.5 | 0.7/0.4 | **2.2/4.9** |
| Guanaco (33B) | **0.1/0.1** | 0.0/0.0 | **0.1**/0.0 |
| Bard-P2 | 4.3/2.3 | 2.9/2.9 | **7.1/15.4** |
| GPT-3.5 | 11.2/10.1 | 6.6/7.6 | **16.6/28.2** |

Table 5: Mean 1SL/5SL EX results when sampling from the train sets, Spider train set and evaluation sets.

### 4.3 Are we truly evaluating the Text-to-SQL datasets in zero-shot or few-shot manner?

We have identified several potential sources of data contamination (Elangovan et al., 2021; Lewis et al., 2021; Magar and Schwartz, 2022) that raise concerns about the true nature of zero-shot or few-shot evaluations of Text-to-SQL datasets. These sources include the availability of both the Spider dataset and classical datasets on GitHub repositories, as well as the presence of the Spider dataset on platforms like Huggingface datasets[5]. Furthermore, the Text-to-SQL datasets may also be included in instruction-tuning dataset collections such as FLAN (Wei et al.). We end the paper with a question for researchers to contemplate: Are we genuinely conducting zero-shot or few-shot evaluations of large language models when they have already been exposed to our evaluation data?

---

[5] https://huggingface.co/datasets/spider

## 5 Related Work

Recently, decoder-based large language models have contributed tremendously to the code-generation tasks (Li et al., 2023b; Fu et al., 2023; Darm et al., 2023). These models leverage unsupervised auto-regressive learning on large-scale text data, allowing them to capture rich semantic relationships and probability distributions of words. Despite their remarkable performance with just one or few-shot examples in context, recent research suggests that they still face challenges on the Text-to-SQL task, which involves complex reasoning (Liu et al., 2023).

There are several works which focus on improving the text-to-SQL parsing capabilities of large language models through enhanced prompt designs. In one study conducted by Nan et al. (2023), the authors emphasize the significance of carefully selecting examples for in-context learning. They demonstrate that incorporating syntactic structures from example queries can greatly enhance the few-shot capabilities of large language models. Chang and Fosler-Lussier (2023) conducted a comprehensive study that explores the impact of prompt length on the performance of text-to-SQL models. Additionally, they examine the sensitivities of representations of database knowledge across various domains. Guo et al. (2023) propose a case-based reasoning framework that adjusts the inputs for GPT-3.5 in cross-domain settings by adaptively retrieving case prompts. Rai et al. (2023) improve the generalization capabilities of large language models with boundary-based techniques that preprocess prompts at both token-level and sequence-level of schema and SQL.

Concurrently, some studies have also explored the potential benefits of complex, multi-step reasoning in improving the performance of large language models on text-to-SQL parsing. Tai et al. (2023) show that Least-to-Most prompting (Zhou et al., 2023) might be unnecessary and directly applying chain-of-thought (CoT) prompts (Wei et al., 2022) could lead to error propagation. Liu and Tan (2023) introduce a divide and prompt paradigm for the Text-to-SQL task, which involves dividing the task into multiple subtasks and applying the CoT approach to each subtask. In another study by Pourreza and Rafiei (2023), a self-correction module is employed in a zero-shot setting to achieve a new state-of-the-art result on the Spider leaderboard. This module feeds the solution of each

sub-problem back to the large language model, enabling it to construct a better overall solution.

# 6 Conclusions and Future Work

This paper systematically evaluates the Text-to-SQL parsing capabilities of six popular large language models across nine benchmark datasets, using five distinct prompt strategies. Our findings indicate that the open-sourced models fall significantly short in performance when compared to closed-source models. However, it is worth noting that even GPT-3.5 performs worse than smaller baseline models on several classical datasets. we are making our outputs available for further analysis and to facilitate future research endeavors. There are several research topics we want to explore in the future. Firstly, we plan to investigate the fine-tuning of these large language models on Text-to-SQL datasets using limited GPU resources using techniques such as low-rank adaptation Hu et al. (2021). Second, we want to explore methods that can dynamically select examples for in-context learning. Third, we would also explore applying model compression techniques to recent Text-to-SQL models (Sun et al., 2023). Lastly, we are interested in examining the feasibility and limitations of employing these large language models on multi-turn Text-to-SQL datasets, such as the SPARC (Yu et al., 2019).

## Limitations

First and foremost, we acknowledge that the scope of this research is limited to six large language models and these models do not encompass the entire research landscape. There have been new and exciting entries to the family, such as the Falcon model.[6] Second, appending five examples to the database schema of some classical datasets might exceed the 2048 token limits of the open-source models in some cases, leading to truncations that might penalize these models with shorter context window. Lastly, some models generate not just SQL statements but also supplementary information, including explanations. To ensure accuracy, we have developed regular expression patterns that aim to extract only the SQL statements to the best of our abilities. Nevertheless, we acknowledge that our rules may not be entirely foolproof and could potentially introduce erroneous SQL in certain cases.

---

[6] https://falconllm.tii.ae/

## Acknowledgments

This research is partially supported by the programme DesCartes funded by the National Research Foundation, Prime Minister's Office, Singapore under its Campus for Research Excellence and Technological Enterprise (CREATE) programme.

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

## A  Appendix

**Prompt *Strategy IS***

A database '**Highschool**' has **3 tables** named 'Highschooler', 'Friend', 'Likes'.
**Highschooler** table has columns: 'ID', 'Name', 'grade'.
**Friend** table has columns: 'student_ID','friend_ID'.
**Likes** table has columns: 'student_ID', 'liked_ID'.

Gave me the SQL query: '**What is Kyle's id**'.
No need explanation.

(a) Examples of prompt strategies IS in the Spider dataset

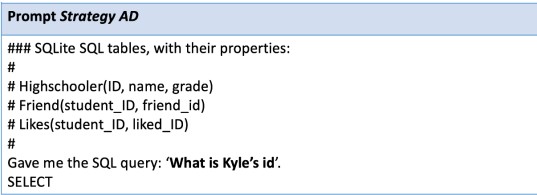

**Prompt *Strategy AD***

### SQLite SQL tables, with their properties:
#
# Highschooler(ID, name, grade)
# Friend(student_ID, friend_id)
# Likes(student_ID, liked_ID)
#
Gave me the SQL query: '**What is Kyle's id**'.
SELECT

(b) Examples of prompt strategies AD in the Spider dataset

**Prompt Strategy Select 3**

```
/*
 3 example rows from table concert:
 SELECT * FROM concert LIMIT 3;
 Table: concert
 concert_ID  concert_Name  Theme  Stadium_ID  Year
 (1, 'Auditions', 'Free choice', '1', '2014')
 (2, 'Super bootcamp', 'Free choice 2', '2', '2014')
 (3, 'Home Visits', 'Bleeding Love', '2', '2015')
*/
---
/*
 3 example rows from table singer:
 SELECT * FROM singer LIMIT 3;
 Table: singer
 Singer_ID  Name  Country  Song_Name  Song_release_year  Age  Is_male
 (1, 'Joe Sharp', 'Netherlands', 'You', '1992', 52, 'F')
 (2, 'Timbaland', 'United States', 'Dangerous', '2008', 32, 'T')
 (3, 'Justin Brown', 'France', 'Hey Oh', '2013', 29, 'T')
*/
-- Using valid SQLite, answer the following question for the tables provided above.
-- How many singers do we have?
SELECT

no need explanation
```

(c) Examples of prompt strategies Select 3 in the Spider dataset

**Prompt Strategy 1SL**

A database '**Highschool**' has **3 tables** named 'Highschooler', 'Friend', 'Likes'.
**Highschooler** table has columns: 'ID', 'Name', 'grade'.
**Friend** table has columns: 'student_ID','friend_ID'.
**Likes** table has columns: 'student_ID', 'liked_ID'.

Translate text to SQL:
How many high schoolers are there? => SELECT count(*) From Highschooler

'**What is Kyle's id**' =>

**Prompt Strategy 5SL**

A database '**Highschool**' has **3 tables** named 'Highschooler', 'Friend', 'Likes'.
**Highschooler** table has columns: 'ID', 'Name', 'grade'.
**Friend** table has columns: 'student_ID','friend_ID'.
**Likes** table has columns: 'student_ID', 'liked_ID'.

Translate text to SQL:
How many high schoolers are there?  => SELECT count(*) From Highschooler
Show the names and grades of each high schooler? => SELECT name, grade FROM Highschooler
Show all the grades of the high schoolers.  => SELECT grade FROM Highschooler
Count the number of high schoolers.  => SELECT count(*) FROM Highschooler
What is the grade of each high schooler?  => SELECT grade FROM Highschooler

'**What is Kyle's id**' =>

(d) Examples of prompt strategies 1SL/5SL in the Spider dataset

Figure 3: Examples of our prompt strategies in the Spider dataset

| Model | #P | PS | Acad | ATIS | Adv | Geo | IMDB | Rest | Sch | Yelp | AVG |
|-------|----|----|------|------|-----|-----|------|------|-----|------|-----|
| Dolly | 3B | IS | 0.5 | 0.3 | 0.0 | 0.5 | 0.0 | 0.0 | 0.0 | 1.6 | 0.4 |
| | | AD | 0.0 | 0.0 | 0.0 | 0.0 | 0.0 | 0.0 | 0.0 | 0.0 | 0.0 |
| | | S3 | 0.0 | 0.0 | 0.0 | 0.0 | 0.0 | 0.0 | 0.0 | 0.0 | 0.0 |
| | | 1SL | 0.0 | 0.3 | 0.0 | 0.0 | 0.0 | 0.0 | 0.0 | 0.0 | 0.0 |
| | | 5SL | 0.0 | 0.0 | 0.0 | 0.0 | 0.0 | 0.0 | 0.0 | 0.0 | 0.0 |
| | 13B | IS | 0.5 | 0.6 | 0.0 | 0.5 | 2.3 | 0.0 | 0.0 | 1.6 | 0.7 |
| | | AD | 0.0 | 0.3 | 0.0 | 0.0 | 0.0 | 0.0 | 0.0 | 0.0 | 0.0 |
| | | S3 | 0.0 | 0.0 | 0.0 | 0.0 | 0.0 | 0.0 | 0.0 | 0.0 | 0.0 |
| | | 1SL | 0.0 | 0.3 | 0.0 | 0.0 | 0.0 | 0.0 | 0.0 | 1.6 | 0.2 |
| | | 5SL | 0.0 | 0.3 | 0.0 | 1.6 | 0.0 | 0.0 | 0.0 | 0.0 | 0.2 |
| | 12B | IS | 0.0 | 0.0 | 0.0 | 8.2 | 3.8 | 0.0 | 0.0 | 3.1 | 1.9 |
| | | AD | 0.0 | 0.0 | 0.0 | 0.0 | 0.0 | 0.0 | 0.0 | 0.0 | 0.0 |
| | | S3 | 0.0 | 0.0 | 0.0 | 0.0 | 1.5 | 0.0 | 0.0 | 0.8 | 0.3 |
| | | 1SL | 0.5 | 0.6 | 0.0 | 8.2 | 0.0 | 0.0 | 0.3 | 2.3 | 1.5 |
| | | 5SL | 0.5 | 0.3 | 0.0 | 1.1 | 0.8 | 0.0 | 0.3 | 0.8 | 0.5 |

Table 6: EX and TS results for Dolly models on classical datasets: **Acad**emic, **ATIS**, **Adv**ising, **Geo**Query, **IMDB**, **Rest**aurant, **Sch**olar and **Yelp**.

| Model | #P | PS | Acad | ATIS | Adv | Geo | IMDB | Rest | Sch | Yelp | AVG |
|-------|-----|-----|------|------|-----|-----|------|------|-----|------|-----|
| Vicuna | | IS | 2.0 | 0.6 | 0.0 | 5.5 | 3.8 | 0.0 | 0.3 | 2.3 | 1.8 |
| | | AD | 0.0 | 0.0 | 0.0 | 2.2 | 0.0 | 0.0 | 0.3 | 1.6 | 0.5 |
| | 7B | S3 | 1.0 | 0.3 | 0.0 | 1.6 | 1.5 | 0.0 | 0.3 | 0.8 | 0.7 |
| | | 1SL | 0.0 | 0.6 | 0.0 | 0.0 | 0.8 | 0.0 | 0.0 | 1.6 | 0.4 |
| | | 5SL | 0.5 | 0.0 | 0.0 | 0.0 | 1.5 | 0.0 | 0.0 | 0.0 | 0.3 |
| | | IS | 2.0 | 0.0 | 0.0 | 0.5 | 6.1 | 0.0 | 0.3 | 2.3 | 1.4 |
| | | AD | 0.0 | 0.0 | 0.0 | 1.6 | 4.6 | 0.0 | 0.3 | 1.6 | 1.0 |
| | 13B | S3 | 1.0 | 0.0 | 0.0 | 4.9 | 3.1 | 0.0 | 1.0 | 2.3 | 1.5 |
| | | 1SL | 0.0 | 0.3 | 0.1 | 4.9 | 0.0 | 0.0 | 0.0 | 0.8 | 0.8 |
| | | 5SL | 0.0 | 0.3 | 0.1 | 2.7 | 0.0 | 0.0 | 0.3 | 0.8 | 0.5 |

Table 7: EX and TS results for Vicuna models on classical datasets: **Acad**emic, **ATIS**, **Adv**ising, **Geo**Query, **IMDB**, **Rest**aurant, **Sch**olar and **Yelp**.

| Model | #P | PS | Acad | ATIS | Adv | Geo | IMDB | Rest | Sch | Yelp | AVG |
|-------|-----|-----|------|------|-----|-----|------|------|-----|------|-----|
| LLaMA | | IS | 0.0 | 0.0 | 0.0 | 0.0 | 0.0 | 0.0 | 0.0 | 0.0 | 0.0 |
| | | AD | 0.0 | 0.0 | 0.0 | 0.0 | 0.0 | 0.0 | 0.0 | 0.0 | 0.0 |
| | 7B | S3 | 0.0 | 0.0 | 0.0 | 3.3 | 0.0 | 0.0 | 0.3 | 0.8 | 0.5 |
| | | 1SL | 0.0 | 0.0 | 0.0 | 0.0 | 0.0 | 0.0 | 0.0 | 0.0 | 0.0 |
| | | 5SL | 0.0 | 0.0 | 0.0 | 0.0 | 0.0 | 0.0 | 0.0 | 0.0 | 0.0 |
| | | IS | 0.0 | 0.0 | 0.0 | 0.0 | 0.8 | 0.0 | 0.0 | 0.8 | 0.2 |
| | | AD | 0.0 | 0.0 | 0.0 | 1.6 | 0.8 | 0.0 | 0.3 | 0.8 | 0.4 |
| | 13B | S3 | 1.0 | 0.6 | 0.0 | 13.2 | 3.1 | 0.0 | 0.3 | 1.6 | 2.5 |
| | | 1SL | 1.0 | 0.0 | 0.0 | 0.0 | 0.0 | 0.0 | 0.0 | 1.6 | 0.3 |
| | | 5SL | 0.0 | 0.0 | 0.0 | 0.0 | 0.0 | 0.0 | 0.0 | 0.8 | 0.1 |
| | | IS | 0.0 | 0.3 | 0.0 | 0.5 | 0.0 | 0.0 | 0.0 | 0.0 | 0.1 |
| | | AD | 0.0 | 0.0 | 0.0 | 12.1 | 3.1 | 0.0 | 0.3 | 1.6 | 2.1 |
| | 30B | S3 | 1.0 | 0.3 | 0.0 | 2.7 | 0.0 | 0.0 | 0.3 | 1.6 | 0.7 |
| | | 1SL | 0.0 | 0.0 | 0.0 | 10.4 | 0.8 | 0.0 | 0.3 | 0.0 | 1.4 |
| | | 5SL | 0.5 | 0.6 | 0.1 | 15.4 | 0.0 | 0.0 | 2.5 | 1.6 | 2.6 |
| | | IS | 0.5 | 0.0 | 0.0 | 1.6 | 0.8 | 0.0 | 0.0 | 2.3 | 0.7 |
| | | AD | 1.0 | 0.0 | 0.0 | 0.5 | 2.3 | 0.0 | 0.0 | 0.0 | 0.5 |
| | 65B | S3 | 0.5 | 0.0 | 0.0 | 6.0 | 2.3 | 0.0 | 0.3 | 0.8 | 1.2 |
| | | 1SL | 0.0 | 0.0 | 0.0 | 0.0 | 0.0 | 0.0 | 0.0 | 0.0 | 0.0 |
| | | 5SL | 0.0 | 0.3 | 0.1 | 11.0 | 0.0 | 0.0 | 2.5 | 0.0 | 1.7 |

Table 8: EX and TS results for LLaMA models on classical datasets: **Acad**emic, **ATIS**, **Adv**ising, **Geo**Query, **IMDB**, **Rest**aurant, **Sch**olar and **Yelp**.