# OpenReview forum: "Battle of the Large Language Models: Dolly vs LLaMA vs Vicuna vs Guanaco vs Bard vs ChatGPT - A Text-to-SQL Parsing Comparison"
_EMNLP/2023/Conference — EMNLP 2023 Findings_

### Official Review · Reviewer_AYeh · 2023-08-01

**Soundness:** 2

**Excitement:**

2: Mediocre: This paper makes marginal contributions (vs non-contemporaneous work), so I would rather not see it in the conference.

**Paper Topic And Main Contributions:**

The authors test six popular large language models, evaluating their Text-to-SQL parsing capability on nine benchmark datasets with five different prompting strategies, covering both zero-shot and few-shot scenarios and found open-source models to be lagging.

**Reasons To Accept:**

The paper appears to be generally well-written.

The authors conduct relatively extensive experiments/analysis.

The authors provide some of the hyperparameters used, which will aid reproducibility.


**Reasons To Reject:**

At least 25 (out of 44 – more than 50%) of the references have no peer-review venue indication or are based on arXiv. This is not good.

The paper’s novelty is very limited, if any. It only compares different LLMs along one single task: text-to-SQL (though over multiple datasets).

Although the objective of the paper is stated, it is not phrased as a research question or in a clear scientific motivation for investigation.

The paper uses only one metric (EX) for evaluation of all models/experiments (except TS for the Spider dataset), thereby limiting the robustness of the results.

The paper does not explain or hypothesize why the anomaly described on line 315 occurs.

It is difficult to know from Table 2 which of the results are EX and which are TS.


**Reproducibility:**

3: Could reproduce the results with some difficulty. The settings of parameters are underspecified or subjectively determined; the training/evaluation data are not widely available.

**Reviewer Confidence:**

4: Quite sure. I tried to check the important points carefully. It's unlikely, though conceivable, that I missed something that should affect my ratings.

**Typos Grammar Style And Presentation Improvements:**

The Introduction explains what text-to-SQL parsing is but without any reference.

The paper can benefit from grammar-check and proof-reading for improvements. For example, line 96.

Unnecessary repetition of some points made in the Section 2.1 about LLMs are repeated in 2.5. Foe example, about the version of ChatGPT used.

---

> ### Author Rebuttal · Authors · 2023-08-28
>
> We appreciate the reviewer for taking the time to read through our manuscript and would like to address some your concerns:
>
> “At least 25 (out of 44 – more than 50%) of the references have no peer-review venue indication or are based on arXiv. This is not good.”:
> We would like to politely point out that the field of LLM develops rapidly and it turns out most of the recent and important work on Text-to-SQL and LLMs are published on arXiv. In this context, we argue it may be unjust to penalize us based on a trend within the NLP landscape.
>
> “The paper’s novelty is very limited, if any. It only compares different LLMs along one single task: text-to-SQL (though over multiple datasets).” and “Although the objective of the paper is stated, it is not phrased as a research question or in a clear scientific motivation for investigation.”:
> Although this paper focuses on Text-to-SQL, we politely argue that this paper can still make a big contribution to this subfield of NLP. We would like to point out that this paper adequately addresses the theme track’s discussion topic on “How reliably do the current generation of LLMs perform on NLP tasks and applications?”.
>
> “The paper does not explain or hypothesize why the anomaly described on line 315 occurs”:
> We believe this can be easily fixed with minor revisions to our manuscript and would like to thank the reviewer for the suggestion.
>
> “It is difficult to know from Table 2 which of the results are EX and which are TS.”:
> TS metric is currently only applicable to the Spider dataset, and Table 2 represents the performance EX scores. We would fix the table caption and apologize for any confusion caused by the unclear description.

---

### Official Review · Reviewer_9kjy · 2023-08-03

**Soundness:** 4

**Excitement:**

4: Strong: This paper deepens the understanding of some phenomenon or lowers the barriers to an existing research direction.

**Justification For Ethical Concerns:**

I do not believe that it is necessary for this paper to go through this process.

**Missing References:**

Satisfactory list of references.

**Paper Topic And Main Contributions:**

The authors cover much of the current landscape when it comes to the Text-to-SQL parsing capability of various large language models, conducting original experiments and demonstrating that, at least in this specific use case, open-source models still produce significantly inferior results when compared to closed-source alternatives.

**Questions For The Authors:**

A: Regarding the final acknowledgement in the "Limitations" section about SQL extraction using regular expressions, is that just a potential problem or an observed one? Has the potential impact on the results been quantified at all?

**Reasons To Accept:**

I find the results interesting as they contradict certain existing assumptions in the community regarding the current performance of open-source LLMs in comparison to closed-source ones.

Even if the results are quickly (or already) not entirely representative of the landscape, I believe it could still be a useful resource in the future due to the quantity of the experimental results on a specific task (Text-to-SQL) and the relevant insights that the authors derive from them.

**Reasons To Reject:**

Due to the nature of the subfield, such papers often quickly become obsolete. In this case, open-source models that could change the conclusions are already available. Falcon was mentioned in the paper itself, and there are claims that the very recently released LLaMA 2 beats GPT-3.5 in various tasks. Perhaps the latter could be added to the paper, assuming the procedure used for testing the original LLaMA is applicable to the newer version.

**Reproducibility:**

5: Could easily reproduce the results.

**Reviewer Confidence:**

4: Quite sure. I tried to check the important points carefully. It's unlikely, though conceivable, that I missed something that should affect my ratings.

**Typos Grammar Style And Presentation Improvements:**

Presentation and language satisfactory.

099. five-->six

---

> ### Author Rebuttal · Authors · 2023-08-28
>
> We would like the reviewer for your recognition and affirmation of our work.
>
> We would like to use this opportunity to address your question “the final acknowledgement in the "Limitations" section about SQL extraction using regular expressions, is that just a potential problem or an observed one? Has the potential impact on the results been quantified at all?”
>
> Response: Figure 1 shows the proportion of valid SQLs we extracted for each prompt strategy/model combination.  We did carefully review the non-sql responses and made sure our regular expressions did not miss out any valid SQL generations. We did not observe any erroneous extraction during our manual review.

---

### Official Review · Reviewer_A359 · 2023-08-11

**Soundness:** 2

**Excitement:**

2: Mediocre: This paper makes marginal contributions (vs non-contemporaneous work), so I would rather not see it in the conference.

**Paper Topic And Main Contributions:**

The author tries to do a comprehensive study for text-to-SQL parsing in terms of different large language models, different prompting strategies and different text-to-SQL datasets. However, the paper lacks more concrete error analysis and deeper reasons which cause different performances among those models. And the novelty of this paper is also limited.

**Questions For The Authors:**

Question A: Are there any different behaviors among the predictions of those LLMs? Is there any tendency for some LLMs to make certain wrong predictions?

**Reasons To Accept:**

The paper provides references for the text-to-SQL parsing performance of several large language models.

**Reasons To Reject:**

The paper reports LLM results for text-to-SQL parsing, but it lacks deeper analysis and insightful findings. In addition, most results look very bad which may indicate additional finetuning or some other strategies like Chain-of-thought is needed. For some LLMs, it may be not suitable to directly do in-context learning for text-to-SQL parsing.

**Reproducibility:**

3: Could reproduce the results with some difficulty. The settings of parameters are underspecified or subjectively determined; the training/evaluation data are not widely available.

**Reviewer Confidence:**

4: Quite sure. I tried to check the important points carefully. It's unlikely, though conceivable, that I missed something that should affect my ratings.

---

> ### Author Rebuttal · Authors · 2023-08-28
>
> We would like to thank the reviewer for reading our manuscript.
>
> We want to politely point out that the goal of this paper is to systematically evaluate representative open-sourced and close-sourced large language models by applying the same set of commonly used zero-shot and few-shot prompting strategies. We argue that it is our controlled experiments which provide important empirical insights, revealing that most of these LLMs are not omnipotent.  The results are important as they can steer the Text-to-SQL community towards potential direction for future research.

---

### Official Review · Reviewer_tPZP · 2023-08-12

**Soundness:** 3

**Excitement:**

2: Mediocre: This paper makes marginal contributions (vs non-contemporaneous work), so I would rather not see it in the conference.

**Paper Topic And Main Contributions:**

Title:
Battle of the Large Language Models: Dolly vs LLaMA vs Vicuna vs Guanaco vs Bard vs ChatGPT - A Text-to-SQL Parsing Comparison

Main contributions:
- Comparison of open source models and closed source models on various benchmarks for text-to-sql parsing.
- Sensitivity to prompt strategies: The effectiveness of different prompting strategies which varies for various models, with no one strategy working for all models.
- Inconsistent Performance Across Models and Datasets: The large language models (LLMs) evaluated in the paper show inconsistent performance across different datasets and prompting strategies. This inconsistency might raise concerns about the generalizability and robustness of the models.
- Few shot learning limitations : Few-shot learning with random examples offered limited performance gains in most cases.

Models Evaluated
Dolly: A 12 billion parameter model, instruction-tuned and licensed for both academic and commercial applications.
LLaMA: A collection of models ranging from 7 billion to 65 billion parameters, trained exclusively on publicly available text corpora.
Vicuna: A 13 billion parameter LLaMA model fine-tuned on user-shared conversations.
Guanaco: A family of models claiming to achieve 99.3% of ChatGPT performance with only 24 hours of fine-tuning.
Bard: A conversational chatbot by Google
GPT-3.5: OpenAI's 175 billion parameter model optimized for chat-based applications.

Evaluation Metrics
The primary evaluation metric is execution accuracy (EX), measuring the alignment of generated SQL queries with gold SQL queries. Test suite accuracy (TS) is also used for the Spider dataset.


**Reasons To Accept:**

The paper offers a detailed and systematic comparison of various language models in the context of Text-to-SQL parsing.

Comprehensive Evaluation of Large Language Models (LLMs): The paper systematically evaluates six popular large language models on their Text-to-SQL parsing capabilities across nine benchmark datasets and five different prompting strategies. This extensive comparison provides valuable insights into the performance of both open-source and closed-source models.

Exploration of Prompting Strategies: The paper investigates various prompting strategies, including zero-shot and few-shot scenarios, to understand their impact on different models. This exploration can provide insights into how different models respond to various prompting techniques, which is a relatively new area of study.

Open source models vs closed source models : The significant performance gap between open-source and closed-source models like GPT-3.5 highlights the need for further research and development in the open-source community. This gap might limit the accessibility and applicability of state-of-the-art models for researchers and practitioners without access to closed-source technologies.

Sensitivity Analysis of Few-Shot Learning: The paper's analysis of how LLMs are highly sensitive to the examples utilized for few-shot learning is an interesting observation that could lead to further research in the area of few-shot learning techniques.

Potential Impact on Text-to-SQL Parsing Research: By making all raw and post-processed outputs publicly available, the paper contributes to the broader research community, facilitating further work in the field of Text-to-SQL parsing.


**Reasons To Reject:**

Lack of innovation : The paper appears to focus solely on text-to-SQL evaluation of existing models without introducing any new techniques or insights.

Lack of Clarity in Methodology: The paper might benefit from a more detailed explanation of the methodology, including the selection criteria for models, the rationale behind the prompting strategies, and the specific challenges faced by the models in different scenarios.

Comparison with State-of-the-Art Models: The paper should include a more comprehensive comparison with state-of-the-art models in Text-to-SQL parsing to provide a clearer context for the performance of the evaluated LLMs.

Challenges with Open-Source Models: The paper notes that open-source models like Vicuna and Dolly face challenges in achieving valid SQL percentage above 75% on classical datasets. But the paper does not provide a thorough analysis of why these models underperform.



**Reproducibility:**

3: Could reproduce the results with some difficulty. The settings of parameters are underspecified or subjectively determined; the training/evaluation data are not widely available.

**Reviewer Confidence:**

5: Positive that my evaluation is correct. I read the paper very carefully and I am very familiar with related work.

---

> ### Author Rebuttal · Authors · 2023-08-28
>
> We really appreciate the reviewer for taking the time to read through our manuscript and would like to address some of the reasons for reject:
>
> “Lack of innovation”: We want to politely point out that this is an empirical paper which directly addresses one of the discussion topics in the theme track: “How reliably do the current generation of LLMs perform on NLP tasks and applications?” We argue that this paper is important to Text-to-SQL practitioners as it systematically examines how the current generation of prompt-based decoder LLMs perform on Text-to-SQL and provides insights on future research directions in this subfield of NLP.
>
> We also want to politely argue that the reasons for rejection stated in “Lack of Clarity in methodology”, “comparison with State-of-the-Art Models” and “challenges with open-source models” can be easily fixed with some minor revisions to our manuscript. We would actively revise our paper to address these issues before the camera-ready deadline.

---

### Meta-Review · Area_Chair_uPF7 · 2023-09-25

**Recommendation:** 3

**Metareview:**

This paper evaluates the text-to-SQL parsing ability of various large language models (LLMs), both open-source and closed-source. The paper compares different LLMs, such as Dolly, LLaMA, Vicuna, Guanaco, Bard, and GPT-3.5, on several text-to-SQL benchmarks, using different prompting strategies. The paper measures the execution accuracy (EX) and test suite accuracy (TS) of the generated SQL queries. The paper finds that open-source models are still far behind closed-source models in text-to-SQL parsing, and that there is no single prompting strategy that works well for all models. The paper also observes that few-shot learning with random examples has limited impact on the performance. The paper provides a comprehensive analysis of the current state-of-the-art and the challenges of text-to-SQL parsing with LLMs.

---

### Decision · Program_Chairs · 2023-10-07

**Decision:**

Accept-Findings

**Comment:**

This paper evaluates the text-to-SQL parsing ability of various large language models (LLMs), both open-source and closed-source. The paper compares different LLMs, such as Dolly, LLaMA, Vicuna, Guanaco, Bard, and GPT-3.5, on several text-to-SQL benchmarks, using different prompting strategies. The paper measures the execution accuracy (EX) and test suite accuracy (TS) of the generated SQL queries. The paper finds that open-source models are still far behind closed-source models in text-to-SQL parsing, and that there is no single prompting strategy that works well for all models. The paper also observes that few-shot learning with random examples has limited impact on the performance. The paper provides a comprehensive analysis of the current state-of-the-art and the challenges of text-to-SQL parsing with LLMs.